# Using Patient-Derived Xenograft (PDX) Models as a ‘Black Box’ to Identify More Applicable Patients for ADP-Ribose Polymerase Inhibitor (PARPi) Treatment in Ovarian Cancer: Searching for Novel Molecular and Clinical Biomarkers and Performing a Prospective Preclinical Trial

**DOI:** 10.3390/cancers14194649

**Published:** 2022-09-24

**Authors:** Jiayu Chen, Yan Li, Haiyuan Wang, Ting Li, Yu Gu, Wei Wang, Ying Shan, Jie Yin, Yongxue Wang, Meng Qin, Siyi Li, Lingya Pan, Siying Peng, Ying Jin

**Affiliations:** 1Department of Obstetrics and Gynecology, Peking Union Medical College Hospital, Chinese Academy of Medical Sciences & Peking Union Medical College, Beijing 100730, China; 2National Clinical Research Center for Obstetric & Gynecologic Diseases, Peking Union Medical College Hospital, Beijing 100730, China; 3BEIJING IDMO Co., Ltd., Beijing 100176, China; 4Precision Scientific (Beijing) Co., Ltd., Beijing 100123, China

**Keywords:** high-grade ovarian cancer, patient-derived xenograft, PARP inhibitor, precise medicine

## Abstract

**Simple Summary:**

The clinical application of PARPis in patients with ovarian cancer has unresolved issues, and whether PARPis can have a similar first-line efficacy to that of platinum-based chemotherapy is still undefined. This study used the PDX model to explore the above problems. We demonstrated that the PDX model can reflect PARPi efficacy more accurately than BRCA mutation, homologous recombination deficiency positivity, and platinum sensitivity. Moreover, the novel clinical and molecular biomarkers suggested that KRAS overexpression was associated with PARPi sensitivity. Additionally, ATK1 enrichment could lead to PARPi resistance, and CA125 less than 10 U/mL during chemotherapy can be a potential indicator for the therapeutic use of PARPi. Above all, PARPis cannot replace platinum-based chemotherapy as first-line treatment in our preclinical trial, indicating that chemotherapy-free tests in the unselected population are not recommended.

**Abstract:**

(1) The accuracy of patient-derived xenografts (PDXs) in predicting ADP-ribose polymerase inhibitor (PARPi) efficacy in ovarian cancer was tested, novel biomarkers were investigated, and whether PARPis could replace platinum-based chemotherapy as a first-line therapy was explored. (2) PDXs were reconstructed for 40 patients with ovarian cancer, and niraparib, olaparib and paclitaxel, and carboplatin (TC) sensitivity tests were conducted. Whole exon sequencing and homologous recombination deficiency (HRD) scores were performed, and patient clinical information was collected. The molecular biomarkers were identified by reverse-transcription quantitative PCR and immunoblotting. (3) Niraparib and olaparib sensitivity were tested in 26 patients and showed high consistency. Approximately half of BRCA wild-type, HRD-negative, and platinum-resistant patients may benefit from PARPis. AKT1 enrichment indicated PARPi resistance; high KRAS expression indicated PARPi sensitivity. CA125 below 10 U/mL during chemotherapy has a sensitivity and specificity similar to platinum sensitivity in predicting PARPi efficacy. Niraparib and TC sensitivity tests were performed on 23 patients, and TC showed a better response in this preclinical trial. (4) PDX can indicate individualized PARPi efficacy. Decreased CA125 levels and KRAS and ATK1 expression levels may be novel biomarkers. The preclinical evidence does not support the implementation of PARPis as the first-line treatment in an unselected population.

## 1. Introduction

Poly ADP-ribose polymerase inhibitors (PARPis) have inspired a new era in epithelial ovarian cancer treatment, with PARPis serving as targeted drugs that competitively inhibit the PARP family, kill tumor cells with homologous recombination deficiency (HRD), and benefit survival [1]. Niraparib and olaparib have been approved by the American Food and Drug Administration (FDA) as advanced post-first-line treatments and maintenance therapies for ovarian cancer, given the significant improvement in patient survival [2,3,4,5,6,7]. China has also recently decided to cover these two medicines with medical insurance for ovarian cancer patients.

Many questions have surfaced with the wide use of PARPis clinically. Most notably, gynecologic oncologists lack evaluation criteria of the clinical efficacy for PARPi maintenance therapy, and the current molecular and clinical indicators for PARPis, such as BRCA1/2 mutations (BRCA1/2muts), homologous repair deficiency (HRD) status, and platinum drug response, lack accuracy [8]. PARPis only benefited two-thirds of relapsed patients with BRCA1/2 mutations [9,10]. The HRD score is a sign of gene scarring. This score can accumulate over time and cannot reflect the real-time appearance of drug resistance, such as that of BRCA recovery mutations [11]. Platinum-sensitive patients have genomic instability due to interstrand crosslink repair deficiency, not HRD [12]. Moreover, wild-type BRCA1/2 (BRCA1/2wt), HRD-negative (HRD–), and platinum-resistant patients can still obtain a benefit from PARPis [13,14,15,16,17,18]. Interestingly, the OReO study demonstrated that BRCA1/2mut and HRD status lost their close relationship with therapeutic effect as PARPi treatment lines increased [19,20].

As a result, some patients have to endure the economic burden and suffer from drug-related adverse reactions, such as bone marrow suppression and digestive system symptoms, without survival benefits. Furthermore, some lose the opportunity for PARPi benefits. Finding a method that more accurately reflects the individual efficacy of PARPis and exploring other clinical and molecular biomarkers are critical for precision treatment with PARPis.

Another phenomenon that caught our attention was the expanding population benefit from PARPis. The Ambition (KGOG 3045) clinical study supported the treatment effect of olaparib in combination with antiangiogenic agents or immunotherapy (PD-L1) on platinum-resistant advanced ovarian cancer with HRD+ [21]. Even more encouraging, preliminary data from a multicenter, prospective, phase 2, single-arm clinical study (NANT) confirmed the efficacy and safety of niraparib monotherapy in neoadjuvant therapy for advanced ovarian cancer patients who bore unresectable lesions [22,23]. These positive results have led us to consider the use of PARPis as a first-line treatment and whether PARPis can replace platinum-based chemotherapy for treating ovarian cancer patients after surgery.

Patient-derived xenografts (PDXs) can simulate tumor development, evolution, and drug response. They are considered one of the best preclinical models because they retain the histopathological, genetic, and tumor microenvironment characteristics of the original tissue [24]. As such, this study re-established PDX models derived from 40 ovarian cancer patients randomly selected from our established PDX library [25] and conducted niraparib, olaparib, and paclitaxel and cisplatin sensitivity tests to evaluate whether PDXs can provide a more accurate personalized efficacy of PARPis, to search for other clinical and molecular markers of PARPis, and to explore the possibility of niraparib as a first-line treatment.

## 2. Materials and Methods

### 2.1. Patients

All patients in this study were from the Department of Obstetrics and Gynecology of Peking Union Medical College Hospital. Informed consent was obtained, and all procedures followed the ethical principles of the Institutional Review Board of Peking Union Medical College Hospital [25]. Clinical information, including PARPi history, therapeutic type (maintenance/treatment), treatment scenario (first-line/second-line), and prognostic information, was updated. Since the median follow-up time was less than two years, the primary endpoint was progression-free survival (PFS), and the secondary endpoint was overall survival (OS).

### 2.2. Establishment of the PDX Models and Performance of Drug Sensitivity Tests

After recovering cryopreserved tumor tissue and implanting a 3 mm × 3 mm × 3 mm tumor mass into female NOD-Prkdcem1Idmo-Il2rgem2Idmo (NPI) mice aged six to eight weeks, we monitored tumor volume and body weight regularly. All procedures were conducted following the ‘Guiding Principles in the Care and Use of Animals’ (China).

Pathological characteristics and the lymphocyte ratio were examined before tissue cryopreservation [25]. All the tumor tissues of PDX models were compared with the original tumor tissues for the consistency of pathology and characteristic proteins. The tumor lymphocyte ratio was detected before cryopreservation of tumor tissues in each PDX model in order to avoid the change of ovarian cancer tumor tissues into lymphoma. Leukocytes were identified as CD19+ and CD45+ (CD19 antibody and PE anti-human CD45 antibody), and samples containing less than 1% leukocytes were qualified for further mouse-to-mouse transplantation.

Then, a drug sensitivity test was performed when the tumor grew to 200 mm^3^. Clinically used olaparib (Lynparza, AstraZeneca AB, Cambridge, UK), niraparib (Zejula, GlaxoSmithKline, Bingford, UK), paclitaxel (Anzatax, Hospira Australia Pty Ltd., Melbourne, Australia), and carboplatin (Paraplatin, MYERS SQUIBB AE, Dubai, United Arab Emirates) were adopted and compared with normal saline (NS), 0.5% methylcellulose (0.5% MC) or 10% HP-β-CD PBS as a vehicle depending on the dissolved reagent.

PDX models generated from the same patient were randomly divided into experimental and control groups (four repeated in each) and received one of the following treatments: (1) niraparib 50 mg/kg (qd) or olaparib 75 mg/kg (qd) via intragastric administration; (2) paclitaxel 30 mg/kg (every four days × 8 cycles) and carboplatin 25 mg/kg (every five days × 6 cycles) via intravenous injection; or (3) the same volume and approach of the corresponding vehicle. Mouse body weight and tumor volume were recorded every three days. The administration was continued until the mice could no longer tolerate the drug, the tumor reached four times the initial volume, or the administration persisted for more than 150 days. Then, tumor tissue was harvested after the mice were euthanized.

### 2.3. Response Evaluation

The best response (%) calculated based on the Modified Response Evaluation Criteria in Solid Tumors (mRECIST) was used to evaluate PARPi efficacy (Appendix A). PFS and OS in the PDXs were defined as the interval between the start of administration and when the tumor had doubled and quadrupled in size, respectively [24]. As long as more than half of the PDX models in the experimental group achieved stable disease (SD), partial response (PR), or complete response (CR) and the best response or survival in the experimental group was better than that in the vehicle group, a patient was considered to respond to PARPis.

### 2.4. WES of Patient Samples and Data Analysis

Tumor and normal tissues were used for whole exon sequencing (WES). The DNA extraction, gene library establishment, and quality control procedures were the same as those in our previous article [25]. Paired tumor and normal tissues were used to detect somatic mutations. The data were analyzed by TCGA-MC3, a scalable open-source scientific approach for mutation calling of tumor exomes using multiple genomic pipelines. All mutations with at least two callers were maintained and annotated by VEP to ensure accuracy, followed by filtering of noncoding regions, mutations of types 2 and 3 in the ClinVar (201912) database, and mutations whose frequencies were more significant than 1% among East Asian populations in the ExAC and gnomAD databases to obtain final somatic mutations. The HaplotypeCaller module of GATK was used to detect germline mutations, which were further filtered by the officially recommended hard-filter criteria of SNP/INDEL and the remaining filter conditions, such as somatic mutations. GSITIC2.0 was used to calculate significantly changed regions (Q-value = 0.01) to determine the copy number variation (CNV) status. DeconstructSigs (R package) based on nonnegative matrix decomposition was employed to extract 96 somatic mutation patterns, map them onto 30 features, and correlate them to data in the COSMIC (version 2) database to predict mutation-driving factors. The HRDscore algorithm, which considers genome heterozygous deletions (LOH), telomere allele imbalances (TAI), and large fragment migrations (LST), was built by Precision Scientific Co. (Beijing).

### 2.5. RNA Extraction and RT-qPCR

Tissue was crushed, and total RNA was extracted with TRIzol reagent (Thermo Fisher Scientific, Inc., Waltham, MA, USA). The mRNA was reverse transcribed into cDNA using the PrimeScript cDNA Synthesis Kit (Takara Bio, Inc., Kusatsu shi, Japan). Applied Biosystems SYBR Green Master Mix (Thermo Fisher Scientific, Inc., US) was employed for qPCR. GAPDH was used as the internal reference. NCBI Primer Blast was used to design the following primers: GAPDH forward 5′-ACCCAGAAGACTGTGGATGG-3′, reverse 5′-TCTAGACGGCAGGTCAGGTC-3′; KRAS forward 5′-ACAGGCTCAGGACTTAGCAA-3′, reverse 5′-AAGGCATCATCAACACCCAGA-3′; ATK1 forward 5′-CTGCACAAACGAGGGGAGTA-3′, reverse 5′-TCACGTTGGTCCACATCCTG-3′.

### 2.6. Protein Extraction and Immunoblotting

Tissue was crushed in RIPA lysis buffer (Beyotime Institute of Biotechnology) with PMSF (1%) (Beyotime Institute of Biotechnology) and a cocktail (1%) (Roche, Basel, Switzerland). This was followed by rotation at 4 °C for 60 min and centrifugation at 12,000 rpm at 4 °C for 15 min. Then, the supernatant was obtained. The protein concentration was determined using a bicinchoninic acid assay kit (LABLEAD). SDS−PAGE loading buffer (Beyotime Institute of Biotechnology) was added to the protein samples, and then the samples were boiled at 100 °C for 10 min. The 25 µg protein sample was subjected to SDS−PAGE electrophoresis (4% concentrated gel and 10% separated gel). Then, it was transferred to a PVDF membrane (EMD Millipore, Burlington, MA, USA), which was ultimately incubated in 5% skim milk at room temperature for one hour. The sealed PVDF membrane was then incubated with the following primary antibodies overnight at 4 °C: rabbit anti-actin monoclonal antibody (mAb) (1:1000; ABclonal, cat. no. A2319), rabbit anti-KRAS polyclonal antibody (pAb) (1:1000; ABclonal, cat. no. A1190) and rabbit anti-AKT1 mAb (1:1000; ABclonal, Wuhan, China, cat. no. A20799). After elution in TBST (LABLEAD), the membrane was incubated in HRP-conjugated goat anti-rabbit IgG (H + L) secondary antibodies (1:5000; ABclonal, cat. no. A5014) at room temperature for one hour. Enhanced chemiluminescence (Thermo Fisher Scientific, Inc., Waltham, MA, USA) was used to detect the signal.

### 2.7. Statistics

Comparisons between the treatment and vehicle groups were performed by Student’s t-tests or Mann−Whitney U tests (measurement data) and chi-squared tests or Fisher’s exact tests (ranked data). The kappa coefficient value was used for consistency analysis, McNemar’s test was used for paired classification data analysis, and the log-rank test and Kaplan–Meier method were used for survival analysis. All data statistics were calculated by SPSS 26.0 (IBM Corporation, Armonk, NY, USA). A *p* value less than 0.05 was considered statistically significant.

## 3. Result

### 3.1. Patient Characteristics and PARPi Sensitivity Test

Forty patients were enrolled, and the detailed clinical information is depicted in Appendix A. All patients except patient 34 (P34) underwent niraparib sensitivity testing (177 PDXs in the experimental group and 181 PDXs in the vehicle group), and olaparib sensitivity testing was performed for 27 patients (109 PDXs in the experimental group and 104 PDXs in the vehicle group). A total of 26 patients underwent both sensitivity tests simultaneously. The experimental group achieved superior drug efficacy compared to the vehicle group. This was indicated by the fact that the niraparib group induced a significantly higher response rate (59.89% vs. 33.15%, *p* < 0.001) and median best response rate (47.34% vs. 109.62%, *p* = 0.001) compared to the vehicle group, and a similar pattern was observed in the olaparib cohort (72.48% vs. 33.65%, *p* < 0.001; 37.06% vs. 113.52%, *p* = 0.001, respectively) (Appendix A).

### 3.2. PDX Is a More Accurate Indicator of the Individualized Efficacy of PARPis

Among the 26 patients who underwent both niraparib and olaparib sensitivity tests, only 4 patients had different efficacy (no significant difference, *p* = 1.00 and high consistency, Kappa index = 0.675, *p* = 0.001) (Figure 1A, Appendix A). In addition, all four patients with clear cell carcinoma (CC) were resistant to PARPis. BRCAmut and HRD+ are relatively reliable indicators of PARPi sensitivity in primary patients with HGSOC, and platinum sensitivity seemed to be a decent indicator in relapsing patients (Appendix A, and tumor volume changes of typical patients are depicted in Appendix A). These findings were consistent with clinical experience and confirmed the repeatability and veracity of PDXs for evaluating PARPi efficacy.

BRCAmut, HRD+, and platinum sensitivity are not accurate indicators of PARPi response since approximately half of BRCAwt, HRD-, and platinum-resistant primary patients respond to PARPis. Additionally, none of these indicators significantly increased the prediction effect for PARPi benefit (Appendix A). The correlation between PARPi efficacy and BRCAmut and HRD+ was even weaker in relapsed patients (Appendix A).

Furthermore, we explored the evaluation efficacy of PDX in patients. Patients who were clinically treated with PARPis as first-line (13 patients) and second-line (9 patients) maintenance therapy were included (Appendix A). They were divided into two groups according to the PARPi response suggested by PDXs. In first-line maintenance patients, the PARPi-effective group had significantly longer PFS than the PARPi-ineffective group (Figure 1B). However, this was not observed in patients with second-line maintenance (Figure 1C) due to a small number of enrolled patients and various second-line treatment regimens. This suggests that PDXs can screen patients with more significant survival benefits from PARPis among the population meeting the existing administration indicators.

### 3.3. Using PDX to Detect Novel Molecular Indicators of PARPis

Patients with WES results and similar sensitivity to olaparib and niraparib were divided into a PARPi-effective group (12 patients) and a PARPi-ineffective group (8 patients), and the genome characteristics of these groups were compared.

#### 3.3.1. CNVs

The effective group showed enriched *Myc* amplification and *APC* deletion (Figure 2A and Appendix A). In addition, *AKT1* deficiency was significantly enriched in the effective group, and *KRAS* deficiency was significantly enriched in the ineffective group (Appendix A). The distribution of CNVs in the two groups was slightly different. The chemotherapeutic resistance-related genes (mainly *AKT1* and *ERBB1*) and *EGFR* tyrosine kinase inhibitor-related genes were decreased in the effective group, and multiple tumor driver genes (mainly *Myc*) were increased in the ineffective group (Figure 2B).

#### 3.3.2. Gene Mutations

The number of functional loss sites in the effective group was significantly increased compared with that in the ineffective group (92.25 ± 51.40 vs. 56.63 ± 20.65, *p = 0.047*) (Appendix A). Then, we analyzed driving mutations in epithelial ovarian cancer. *TP53* mutations were the most common (63%), followed by *NF1* (20%) and *NF2* (10%) mutations. Additionally, these mutations were enriched in the effective group in general, especially *BRCA1/2* mutations (Figure 2C). Nevertheless, no significant difference in gene mutations was detected (Appendix A). In the signature analysis, only age was identified as significantly different between the effective group (lower) and the ineffective group (higher) (Appendix A). Mutual exclusion and coexistence examination of mutated genes highlighted a coexisting trend of *BRCA1*-*LRP1B* and *NOTCH1*-*LATS1* mutations in the effective group, but the differences were not significant (Appendix A).

#### 3.3.3. Verification by RNA and Protein

Patients in the niraparib responder (P08, P15, P24) and nonresponder (P17, P23, P32) groups were randomly selected to explore the effects of KRAS and AKT1 on PARPi efficacy. In both the pre-niraparib and post-niraparib tissues, the RNA levels of *KRAS* in the responder group were significantly higher than those in the nonresponder group, but *AKT1* showed no significant difference (before treatment, *p* = 0.618; after treatment, *p* = 0.389) (Figure 2D,E). Therefore, we further explored the change in AKT1 expression after treatment. After niraparib treatment, the AKT1 level showed an increasing trend, although there was no statistical significance (Figure 2F). The protein analysis showed similar results. The KRAS level before treatment in the responder group was significantly higher than that in the nonresponder group, but there was no significant difference after treatment (*p* = 0.402) or in the AKT1 level (before treatment, *p* = 0.125; after treatment, *p* = 0.055) (Figure 2G–I). Similarly, AKT1 levels significantly increased after niraparib treatment, while KRAS levels significantly decreased after niraparib treatment in the responder group (Figure 2J,K).

These experiments suggest that tumor tissues with high KRAS expression are responsive to niraparib and that the enrichment of AKT1 during treatment might lead to drug resistance.

### 3.4. Using PDX to Detect Novel Clinical Indicators of PARPis

Platinum sensitivity is the clinical indicator for niraparib when it is used as maintenance therapy. However, it takes at least six months to determine the platinum response in naive patients, which can delay the administration. The degree of CA125 reduction during chemotherapy may partly reflect the sensitivity to platinum-based chemotherapy. As a result, this study investigated whether the lowest CA125 level (truncated at 10 U/mL) during chemotherapy could replace platinum sensitivity as a predictor of PARPi efficacy. Seventeen out of twenty-two patients with CA125 levels of no more than 10 U/mL were platinum-sensitive, while 5 out of 15 patients with CA125 levels less than 10 U/mL were platinum-resistant. The sensitivity and specificity of CA125 for predicting the efficacy of PARPis were 77.27% and 66.67% (Figure 3A), respectively, while those of platinum sensitivity were 86.36% and 40.00% (Figure 3B), respectively. These values were not significantly different (*p* = 0.687, *p* = 0.289, respectively) (Figure 3C,D).

### 3.5. Using PDX to Perform Clinical Trials

#### 3.5.1. Reproducing NOVA, PRIMA, and SOLO I Trials

PDXs from five platinum-sensitive relapsing patients with HGSOC were employed to mimic the NOVA study. Both the PFS and OS of the niraparib group were significantly better than those of the control group (Appendix A). Five platinum-sensitive primary patients with BRCAmut were employed to mimic the SOLO I study, and all PDXs in the experimental group had a longer PFS and OS than those in the vehicle group (Appendix A). Seven platinum-sensitive advanced patients with residual lesions after surgery were employed to imitate the PRIMA study, including two patients with HRD+ and one with BRCAmut. All PDXs, HRD+ PDXs, and HRD- PDXs benefited from niraparib, as did the small number of BRCAmut patients who could not be analyzed effectively (Appendix A).

#### 3.5.2. Attempting a Prospective Trial to Determine Whether Niraparib Could Replace Chemotherapy as First-Line Treatment for Ovarian Cancer

We further recruited PDXs established by 23 naive patients and compared the efficacy of the vehicle, niraparib, and TC chemotherapy. The efficacy evaluation indexes included best response (%), response category rate, PFS, and OS. The TC group exhibited a better treatment effect than the niraparib group among the general population (Figure 4A,B and Appendix A) and patients in the HRD+ or BRCAmut subgroups (Figure 4C–F and Appendix A). We noticed that 11 patients responded to both niraparib and TC (Figure 4G). Thus, we further explored this possibility in these patients and observed the same results (Figure 4H,I and Appendix A). Nevertheless, niraparib and TC demonstrated a similar efficacy in several patients (P08 and P15) (Figure 4J), and two TC-resistant patients responded to niraparib (Figure 4G).

Therefore, PDX can reproduce the clinical trials well, and niraparib has not yet been indicated as a replacement for TC chemotherapy in the first-line treatment of ovarian cancer in our preclinical trial.

## 4. Discussion

PDXs are considered one of the best preclinical models. In PDXs, the original tumor tissue is subcutaneously transplanted into immunodeficient mice, and PDXs retain their molecular and pathological characteristics, drug response, and tumor microenvironment despite host selection pressure [24,26,27,28]. The samples in the PDX model library used in this paper have been compared with the original tumor genome, transcriptome, proteome, and chemotherapy response data and have shown good consistency [25].

PDXs have been used to screen small-molecule targeted drugs for various tumors, and HGSOC-PDXs have been used to explore the PARPi response of patients with RAD51 or BRCA1 methylation and the therapeutic effect of PARPis combined with other drugs [29,30,31]. However, whether PDXs can accurately reflect PARPi efficacy has not been clearly indicated in experiments because few studies have reported consistent results on the correlation between BRCAmut and olaparib sensitivity [24,26]. In this study, the niraparib and olaparib sensitivity tests showed high consistency in PDXs established from the same patient, suggesting the repeatability of the PDX model for evaluating PARPi efficacy. Naive HGSOC patients with BRCAmut and HRD+ were more responsive to PARPis than platinum sensitivity. Additionally, platinum sensitivity seemed to be a more reliable indicator in relapsed patients, as identified by clinical experience and trials. Moreover, the PDX model reproduced the NOVA, PRIMA, and SOLO I trials. These results suggest that the PDX model can accurately predict the PARPi response.

The most prominent problem with PARPis in ovarian cancer is the lack of adequate application metrics, and BRCA mutation, HRD status, and platinum sensitivity are not perfect indicators for PARPis [10,14,15,16,17,32,33]. The mechanism of PARPis can partially explain this phenomenon. Although PARPis can lead to a ‘synthetic lethal’ interaction with HRD, the compensatory effects of other DNA damage repair pathways, BRCA1/2 recovery mutations, high expression of drug resistance pumps, and changes in tumor metabolism may cause acquired PARPi resistance [34]. In addition, other molecular defects in the homologous recombination (HR) pathway have the same killing effect (called ‘BRCAness’) [35]. Platinum agents lead to chromosomal cross-linking and kill tumor cells once nucleotide excision repair deficiency exists, indicating the presence of genomic instability and not HRD [12]. Therefore, we tested the accuracy of the current indicators and explored novel molecular and clinical indicators using PDXs.

In this experiment, we found that 20% of HRD+ and 40% of platinum-sensitive naive HGSOC patients had no response to PARPis, and approximately half of BRCAwt, HRD-, and platinum-resistant HGSOC patients responded to PARPis. In addition, we screened subpopulations with more significant survival benefits from patients clinically treated with PARPis as first-line maintenance therapy, demonstrating that the PDX model can effectively avoid the above limitations of clinical and molecular indications. Thus, the PDX model serves as a ‘black box’ that provides a more accurate and objective efficacy prediction than BRCAmut, HRD+, and platinum sensitivity.

PDX models feature patient-specific therapeutic effects and thus provide a possible tool for searching for molecular and clinical biomarkers of PARPis. Our research suggested that tumors with high KRAS expression were sensitive to PARPi treatment, and increased AKT1 expression after PARPi treatment may be associated with PARPi resistance.

The oncogene KRAS is associated with tumor development and drug resistance. Research has revealed that MEK inhibitors obstructing abnormally activated KRAS signaling pathways could trigger and amplify PARPi efficacy by increasing double-strand DNA breaks and activating the STING signaling pathway [36]. Although TP53 mutations are predominant in high-grade serous carcinoma and are the most common pathological type of epithelial ovarian cancer, KRAS mutations are more common in low-grade serous carcinoma and endometriosis-driven pathological types (clear cell or endometrioid carcinoma). PARPis are mainly used in the treatment of HGSOC patients, given that BRCA1/2 mutations mainly exist in HGSOC. The relationship between the KRAS expression level and PARPi efficacy in this study may provide a new direction for further preclinical trials of PARPi usage in such rare ovarian cancers and other malignancies with predominant KRAS mutations, such as colon cancer [37,38].

Through WES, ATK1 deletion mutations were enriched in the PARPi-sensitive group. We found that ATK1 was significantly enriched in tumors after PARPi application, and the residual tumor cells after PARPi administration were mostly drug-resistant cells after drug screening, which indicated that AKT1 enrichment might be related to PARPi resistance. However, there was no difference in AKT1 between the sensitive group and the drug resistance group, which may be caused by the insufficient number of tissues included. Additionally, PARPis activate AKT to form the phosphorylated ATM-Nemo-Akt-MTOR complex, located in the mitochondria, to protect cells from oxidative stress [39]. These two studies are consistent with our results, but further studies need to explore the underlying mechanisms in more detail.

Platinum sensitivity is the clinical indicator for PARPis, and the decrease in CA125 is closely correlated with platinum reactivity [8,40,41]. Therefore, adopting 10 U/mL as the cutoff value, this study showed that the minimum level of CA125 during chemotherapy could replace the platinum response as a predictor for the use of PARPis given that the two methods had similar sensitivity and specificity. This suggests that patients with CA125 less than 10 U/mL during chemotherapy can use PARPis as maintenance therapy immediately.

The other aspect that caught our attention was that PARPis have continuously expanded their application in ovarian cancer treatment, from posterior-line treatment to second-line maintenance to the current first-line maintenance [2,3,4,5,6,7]. Moreover, a study supported niraparib monotherapy as a neoadjuvant agent in advanced ovarian patients who cannot undergo optimal cytoreductive surgery [22,23]. Given this, we questioned whether PARPis had a similar first-line treatment effect to that of platinum-based chemotherapy in the PDX model. Unfortunately, TC consistently outperformed niraparib. However, it is undeniable that niraparib has shown similar efficacy to TC in several patients, and TC-resistant patients may benefit from niraparib. However, there is no approach to distinguish this subpopulation except the PDX susceptibility test. Therefore, clinical trials should be conducted with caution.

A limitation of this study is the small number of included patients. This made it challenging to analyze the molecular targets related to PARPi efficacy and identify subgroups that could benefit from niraparib as first-line treatment. The long time period that is required for model formation and drug sensitivity testing is often listed as a limitation to clinical application. However, the median PDX formation time was five months, and the median drug sensitivity testing time for patients lacking sensitivity (no effects on OS) was 23 days. That is, the PARPi maintenance efficacy results could be obtained within six months, which would not affect patient use of PARPis.

## 5. Conclusions

PDX models can better represent the personal therapeutic efficacy of PARPis in epithelial ovarian cancer than BRCA1/2mut, HRD+, and platinum sensitivity. High KRAS expression was correlated with PARPi sensitivity, and the accumulation of AKT1 during PARPi treatment might lead to PARPi resistance. A minimum CA125 of less than 10 U/mL during chemotherapy can be used as a clinical indicator of PARPi sensitivity. PARPis are not yet an alternative to platinum-based chemotherapy as a first-line treatment for ovarian cancer.

## Figures and Tables

**Figure 1 cancers-14-04649-f001:**
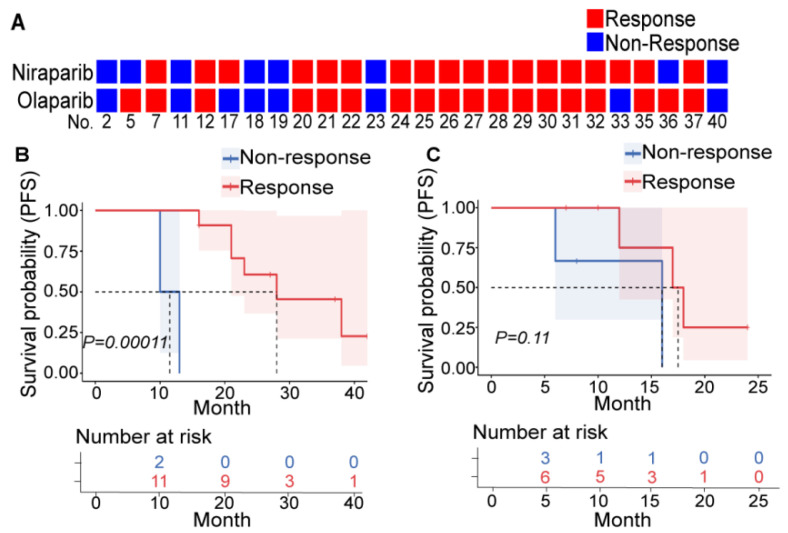
PDXs can reflect PARPi efficacy. (**A**) Results of the niraparib (upper row) and olaparib (lower row) susceptibility tests in PDXs derived from 26 patients. Each square represents a response to a drug (red: nonresponder; blue: responder), and each column represents a patient (the patient number is at the bottom). Patients who were treated with PARPis as first-line (**B**) or second-line (**C**) maintenance therapy were divided into the responder (red) and nonresponder (blue) groups according to the PDX drug sensitivity test, and the differences in PFS between these two groups were compared. The upper picture depicts the Kaplan–Meier curve and log-rank value between the responder and nonresponder groups. The lower table shows the number of patients who did not reach the endpoint at the corresponding time points in each group.

**Figure 2 cancers-14-04649-f002:**
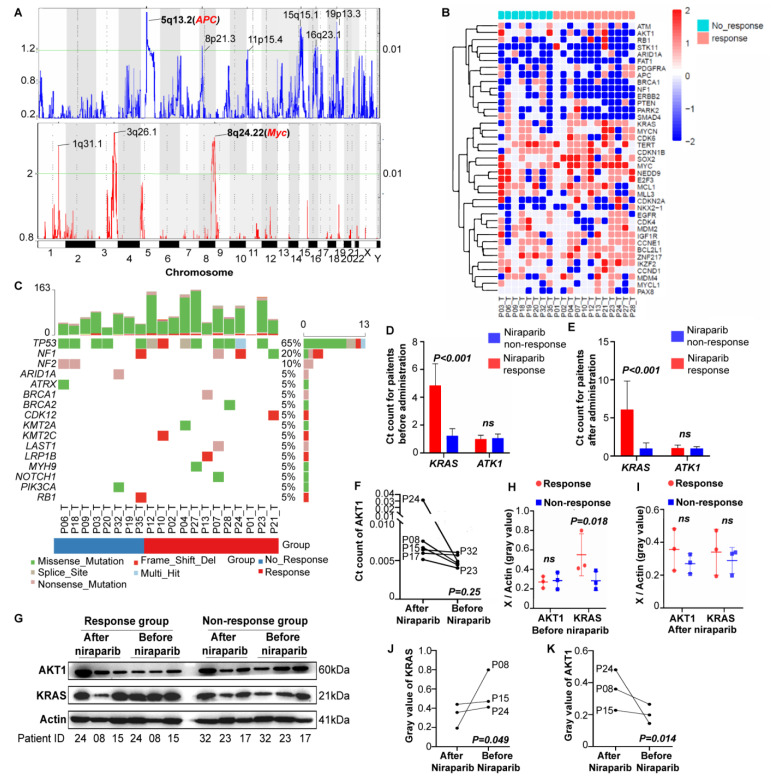
Finding novel molecular biomarkers of PARPis. (**A**) The copy number variations (CNVs) in the olaparib and niraparib response groups. The bottom of the figure indicates the chromosome, the left number is the GISTIC Gscore (CNV frequency multiplied by CNV amplitude), and the right number is the q-value. The upper figure shows gene deletions (blue), and the lower figure shows gene enrichments (red). The detailed loci of any statistically significant CNVs (*p* < 0.01) are labeled. (**B**) The cluster analysis of CNVs in the responder and nonresponder groups. (**C**) Driver mutation analysis in the responder and nonresponder groups. The RNA (**D**–**F**) and protein (**G**–**K**) levels of KRAS and AKT1 were measured in the PDX tumor tissue before and after niraparib administration in the responder and nonresponder groups (three in each). Differences in KRAS and AKT1 RNA levels between the responder and nonresponder groups before (**A**) and after (**B**) niraparib treatment. (**C**) Changes in AKT1 RNA levels before and after niraparib treatment in the same patient. (**D**) The immunoblotting result, followed by the analysis of its gray values. Differences in KRas and Akt1 protein levels between the responder and nonresponder groups before (**E**) and after (**F**) niraparib treatment. Changes in Akt1 (**G**) and KRas (**H**) levels before and after niraparib treatment in the same patient from the responder group.

**Figure 3 cancers-14-04649-f003:**
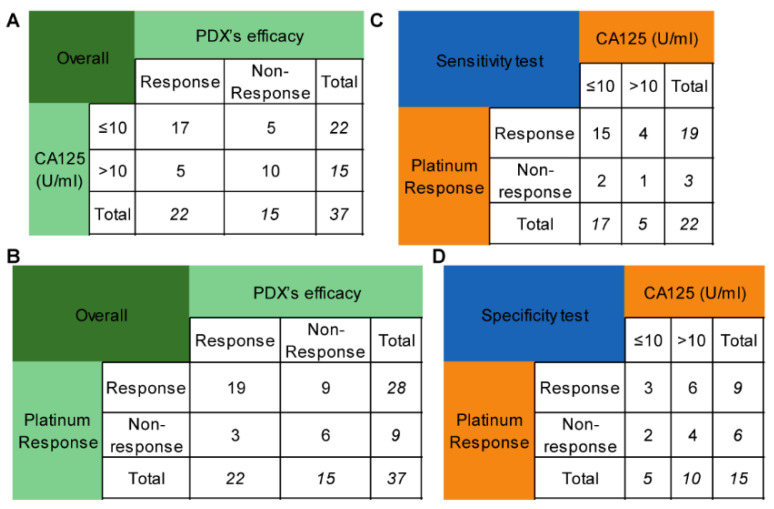
Comparison of the CA125 level and platinum response in predicting niraparib efficacy. (**A**) The performance of the CA125 level (truncated at 10 U/mL) in predicting niraparib efficacy. (**B**) The performance of the platinum response in predicting niraparib efficacy. (**C**) Comparison of the sensitivity of the CA125 level and platinum response to predict niraparib efficacy. (**D**) Comparison of the specificity of the CA125 level and platinum response to predict niraparib efficacy.

**Figure 4 cancers-14-04649-f004:**
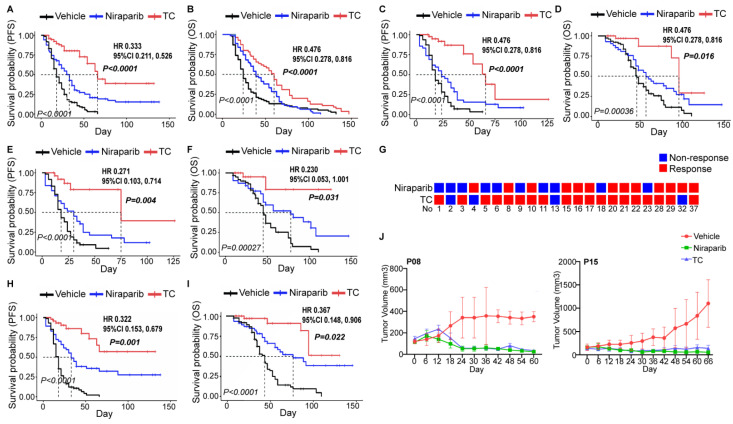
Comparing the efficacies of niraparib and TC in PDX. The survival outcomes of the control, niraparib, and TC chemotherapy groups in PDXs of naive patients (**A**,**B**), the HRD+ subgroup (**C**,**D**), or the BRCA mutation subgroup (**E**,**F**) were analyzed. (**G**) Results of the niraparib and TC susceptibility tests evaluated by PDXs in 23 patients. (**H**,**I**). The survival outcomes of the control, niraparib, and TC chemotherapy groups in patients who simultaneously responded to niraparib and TC. (**J**) Tumor volume changes due to the vehicle (red), niraparib (green), and TC (blue) treatments in 2 patients (P08 and P15) who were responsive to niraparib but not TC. The italic *p* value in each Kaplan–Meier survival curve figure indicates the comparison of those three groups, and the *p* value (bold and italic), hazard ratio (HR), and 95% confidence interval (CI) represent the comparison between the niraparib and TC groups. TC, paclitaxel and carboplatin chemotherapy; PFS, progression-free survival; OS, overall survival.

## Data Availability

All data generated or analyzed during this study are included in this published article and its Appendix A.

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
