# Peer review of "Using Patient-Derived Xenograft (PDX) Models as a ‘Black Box’ to Identify More Applicable Patients for ADP-Ribose Polymerase Inhibitor (PARPi) Treatment in Ovarian Cancer: Searching for Novel Molecular and Clinical Biomarkers and Performing a Prospective Preclinical Trial"

_cancers, 2022, doi:10.3390/cancers14194649_

Round 1

Reviewer 1 Report

Comments:

1.      The authors have not specified a subtype of ovarian cancer in their work. Different ovarian tumor types (e.g. high-grade serous ovarian cancer or clear cell carcinoma of the ovary) are significantly distinct in many aspects including molecular profile, disease prognosis, treatment regimens, and overall disease management. The Authors should indicate ovarian cancer subtype when describing patients’ population or PDX models in this manuscript.

2.      KRAS is very rarely mutated in ovarian cancer when compared with other tumor types. For instance, KRAS mutation is a well-known predictive biomarker of resistance to EGFR-I-based therapy in colon cancer. In epithelial ovarian cancer, however, KRAS is very rarely mutated (~0.5% of all cases). In this work, the Authors used mRNA and protein levels of KRAS to predict responsiveness to PARPi, which could be challenging to determine what expression levels of KRAS are considered to be increased and qualify for PARPi. The Authors should address this issue in the discussion how mRNA or protein levels of KRAS could be used routinely in the clinic as a predictive biomarker for ovarian cancer patients?

Typos:

Line 25: can not

Line 29 norvel

Author Response

  1. The authors have not specified a subtype of ovarian cancer in their work. Different ovarian tumor types (e.g. high-grade serous ovarian cancer or clear cell carcinoma of the ovary) are significantly distinct in many aspects including molecular profile, disease prognosis, treatment regimens, and overall disease management. The Authors should indicate ovarian cancer subtype when describing patients’ population or PDX models in this manuscript. 

Answer: Thanks for your suggestion. The clinical information of patients is depicted in Table 1, and we are sorry that the tables fail to be uploaded. We have uploaded this table as an attachment (called "Table"), in which the histopathological types are described in detail.

  1. KRAS is very rarely mutated in ovarian cancer when compared with other tumor types. For instance, KRAS mutation is a well-known predictive biomarker of resistance to EGFR-I-based therapy in colon cancer. In epithelial ovarian cancer, however, KRAS is very rarely mutated (~0.5% of all cases). In this work, the Authors used mRNA and protein levels of KRAS to predict responsiveness to PARPi, which could be challenging to determine what expression levels of KRAS are considered to be increased and qualify for PARPi. The Authors should address this issue in the discussion how mRNA or protein levels of KRAS could be used routinely in the clinic as a predictive biomarker for ovarian cancer patients?  

Answer: Thanks for your suggestion, and we have discussed this issue in the discussion part(Line 410-417) - ”Although TP53 mutations are predominant in high-grade serous carcinoma, the most common pathological types of epithelial ovarian cancer, KRAS mutations are more common in low-grade serous carcinoma and endometriosis-driven pathological types (clear cell or endometrioid carcinoma). PARPi is mainly used in the treatment of HGSOC patients, given that BRCA1/2 mutations mainly exist in HGSOC. The relationship between KRAS expression level and PARPi efficacy in this study may create a new direction for the further preclinical trial of PARPi usage in such rare ovarian cancer and other malignancies with predominant KRAS mutations, such as colon cancer.”

Reviewer 2 Report

The authors established patient-derived xenografts (PDXs) models derived from forty ovarian cancer patients and examined PDXs providing more accurate personalized efficacy of PARPi based on whole exon sequencing. Niraparib and olaparib susceptibility tests in PDXs derived from 26 patients showed PDX reflecting the PARPi efficacy. Tumor tissues with high KRAS expression were responsive to niraparib; high expression AKT1 during treatment might lead to drug resistance. These results show helpful for clinical application. It provides the potential for further research. This full article constructed adequate background in introduction section and detailed methods. The researchers achieve valuable clinical information. I suggest accept in the present form.

Author Response

The authors established patient-derived xenografts (PDXs) models derived from forty ovarian cancer patients and examined PDXs providing more accurate personalized efficacy of PARPi based on whole exon sequencing. Niraparib and olaparib susceptibility tests in PDXs derived from 26 patients showed PDX reflecting the PARPi efficacy. Tumor tissues with high KRAS expression were responsive to niraparib; high expression AKT1 during treatment might lead to drug resistance. These results show helpful for clinical application. It provides the potential for further research. This full article constructed adequate background in introduction section and detailed methods. The researchers achieve valuable clinical information. I suggest accept in the present form. 

Answer: Thank you for your recognition of our work, and thank you for your hard work in reviewing the manuscript.

Reviewer 3 Report

The manuscript was aimed to show the benifit to use the patient-derived xenografts to predict the efficacy of PARP inhibitors (PARPi)  for therapy of  patients with ovarian cancer (OC) and to compare its accuracy with the other markers/assays commonly used before (e.g. BRCA status, homologous recombination deficiency (HRD+), etc.)). 

I have the following concerns and suggestions about the manuscript.

Some of the important material is missing in the manuscript.

1) The characteristics of the patients declared to be presented in the Table 1  (line 190). However, manuscript lack this Table;

2) Chapter "Materials and methods" lack the description of the assay(s) used to assess the efficiency of homologous recombination to show homologous recombination deficiency (HRD) in OC;

3) The role of lymphocyte ratio (line 107) in PARPi sensitivity was not examined in the manuscript;

4) Chapter 3.1 lacks any word about the experimental group 2, received chemotherapy (paclitaxel and carboplatin).

5) The histopathological examination of PDXs is highly desirable to define partical and complete responses, but this assay is also missing in the manuscript. 

6) Given that the authors argue about HRD status in OC to be not accurate to assess sensitivity to PARPi, it looks surprising that expression of Rad51 recombinase was not assessed in tumors and PDX.  

7) The authors argue that AKT1 deficiency was significantly enriched in the effective group (line 239) and thereby can be used as a marker of resistance to PARPi (lines 24, 37 etc.). However,  WB data shown in Figure 2G did not support this notice. Similarly, the normalized data shown in Figure 2H did not support this argument.  

8) The authors also have to discuss the potential molecular mechanisms to show how high level of KRAS expression in OC involved in sensitivity of cancer cells to PARPi. 

9) An extensive language editing is required for this manuscript.  

Author Response

1)The characteristics of the patients declared to be presented in the Table 1  (line 190). However, manuscript lack this Table;

Answer: Thanks for your suggestion. The clinical information of patients is depicted in Table 1, and we are sorry that the tables fail to be uploaded. We have added this table at the end of the manuscript, in which the histopathological types are described in detail.

2) Chapter "Materials and methods" lack the description of the assay(s) used to assess the efficiency of homologous recombination to show homologous recombination deficiency (HRD) in OC;

Answer: Thanks for your suggestion. HDR score was detected and analyzed by the test kit of Precision Scientific (Beijing) CO., LTD. All rights of interpretation are reserved by this company.

3) The role of lymphocyte ratio (line 107) in PARPi sensitivity was not examined in the manuscript;

Answer: The tumor lymphocyte ratio was detected before the cryopreservation of tumor tissues in each PDX model in order to avoid the change of ovarian cancer tumor tissues into lymphoma. Specific operations are as follows: CD19 antibody and PE anti-human CD45 antibody. Leukocytes were identified as CD19+ and CD45+, and samples containing less than 1% leukocytes were qualified for further mouse-to-mouse transplantation. In this paper, the established PDX model library is used for further tests, and the data can be found in our previous post (“Chen JY, Jin Y, Li SY, Qiao C, Peng XX, Li Y, et al. Patient-Derived Xenografts Are a Reliable Preclinical Model for the Personalized Treatment of Epithelial Ovarian Cancer. Frontiers in oncology 2021;11.”).

4) Chapter 3.1 lacks any word about the experimental group 2, received chemotherapy (paclitaxel and carboplatin).

Answer: Thanks for your question. Chapter 3.1 mainly described the main results of PARPi susceptibility in the PDX model as the subtitle suggested “Patients’ characteristics and PARPi sensitivity test”. PDXs receiving chemotherapy (paclitaxel and carboplatin) sensitivity tests were written in Chapter 3.5.2 (line 337,338). The drug sensitivity results are shown in Table S2, and the specific information is described in Original Materials Table 4.

5) The histopathological examination of PDXs is highly desirable to define partical and complete responses, but this assay is also missing in the manuscript. 

Answer: All the tumor tissues of PDX models were compared with the original tumor tissues for the consistency of pathology and characteristic proteins. In this paper, the established PDX model library is used for further tests, and the data can be found in our previous post (“Chen JY, Jin Y, Li SY, Qiao C, Peng XX, Li Y, et al. Patient-Derived Xenografts Are a Reliable Preclinical Model for the Personalized Treatment of Epithelial Ovarian Cancer. Frontiers in oncology 2021;11.”).

6) Given that the authors argue about HRD status in OC to be not accurate to assess sensitivity to PARPi, it looks surprising that expression of Rad51 recombinase was not assessed in tumors and PDX.  

Answer: Thanks for your question. Although Rad51 recombinase indicates homologous recombinase repair defects, the BRCA1/2 mutation, HRD score, and platinum sensitivity are recommended by the NCCN guidelines for the clinical application of PARPi. This paper is based on the problems faced in the clinical application of PARPi. Perhaps further we'll re-evaluate Rad51 recombinase's link with the PARPi application.

7) The authors argue that AKT1 deficiency was significantly enriched in the effective group (line 239) and thereby can be used as a marker of resistance to PARPi (lines 24, 37 etc.). However,  WB data shown in Figure 2G did not support this notice. Similarly, the normalized data shown in Figure 2H did not support this argument.  

Answer: Thanks for your question. Through WES, we found that ATK1 deletion mutations were enriched in the PARPI-sensitive group. We found that ATK1 was significantly enriched in tumors after PARPi application, and the residual tumor cells after PARPi administration were mostly drug-resistant cells after drug screening, which indicated that AKT1 enrichment might be related to PARPi resistance. However, there was no difference in AKT1 between the sensitive group and the drug resistance group, which may be caused by the insufficient number of tissues included.

8) The authors also have to discuss the potential molecular mechanisms to show how high level of KRAS expression in OC involved in sensitivity of cancer cells to PARPi. 

Answer: Thanks for your questions. The aim of this paper is to explore new molecular targets for the efficacy of PARPi. Further mechanistic studies are underway and will be the focus of our next research. Thank you for providing us with new directions for future research.

9) An extensive language editing is required for this manuscript.  

Answer: Thanks for your suggestions. We embellished the language by American Journal Experts@, and polished proof has been provided.

Round 2

Reviewer 3 Report

The authors responded to majority of the suggestions. The manuscript can be accepted for publication. 

Author Response

The authors responded to majority of the suggestions. The manuscript can be accepted for publication. 

Answer: Thank you for your comments and suggestions on the article.